# Synchronization, Attention and Transformation: Multidimensional Exploration of the Aesthetic Experience of Contemporary Dance Spectators

**DOI:** 10.3390/bs8020024

**Published:** 2018-02-10

**Authors:** Coline Joufflineau, Coralie Vincent, Asaf Bachrach

**Affiliations:** 1UMR 8218, Institut ACTE, 75015 Paris, France; 2UMR 7023 CNRS/Paris 8, 75017 Paris, France; coralie.vincet@cnrs.fr (C.V.); asaf.bachrach@cnrs.fr (A.B.); 3ICI-project, Labex Arts H2H, Université Paris 8, 93526 Saint-Denis, France

**Keywords:** dance spectating, time perception, interpersonal resonance, attention, shared present

## Abstract

The co-presence of bodies in intersubjective situations can give rise to processes of kinesthetic empathy and physiological synchronization, especially in the context of dance: the body and attention of the spectators are oriented towards the dancers. In this study, we investigate the processes of “body-mind” resonance between a choreography and its spectators, and more specifically the lasting impact of this resonance post-performance. We then explore the relation between the observed effects and subjective measures of attention. The study focuses on the work of the French choreographer Myriam Gourfink, who develops a unique movement, based on the slower breathing of dancers: the breathing generates an extremely slow movement without rhythmic ruptures. Phenomenological studies of her work report changes in temporal perception and changes in bodily attentional states. We made use of two cognitive tasks in order to quantify this change in temporal perception: Spontaneous Motor Tempo (SMT) and Apparent Motion effect (AM) before and after a 40-min live performance. Subjective reports were collected at the end of the performance. Physiological data were recorded before and after the performance. We performed a control experiment with a choreography of a distinctly different quality of movement. Post-Gourfink performance, we observed a significant deceleration of SMT and a decrease in its variability, while AM was reported with longer temporal intervals. Neither of these effects was observed in the control condition. Furthermore, an increase in perception of AM was correlated with a slower breathing rate after the performance. Correlations with subjective reports suggest a link between changes in cognitive and physiological dynamics and the degree of absorption of the spectators in the performance. In addition, these changes were related to specific reported attentional dispositions that we interpret as a form of attentional resonance. The ensemble of the results suggests an expansion of the “specious present” that is related to the slowing of physiological rhythms, and an attentional resonance between spectators and the choreography. The intricate relation we observed between inter-personal resonance and temporal cognition, foregrounds the notion of shared present as a neurophenomenological construct.

## 1. Introduction

The concept of resonance (broadly understood as processes of synchronisation, including in-phase or phase-delayed behaviour, co-variation of gestures, facial or vocal expression and co-variation of physiological rhythms and affective dynamics [1]) has a long history in the neurocognitive study of social interactions [2]. It has also been evoked in the more recent field of neuroaesthetics, studying artistic production/reception [3], and in particular in studies of the perception and the production of music [4,5]. In the study of social interactions, which explores the phenomena of interpersonal resonances in dyads and groups, these echo phenomena have been described or quantified in various scales: neural, physiological, sensorimotor, behavioral, and experiential or affective. Thus, different studies show inter-individual neural synchronizations in situations of joint attention towards an external object or when individuals look into each other’s eyes [6], as well as in different forms of joint action such as mimicry [7]. At the motor and behavioral level, several studies show that coordinated motor responses appear without any explicit instruction to the participants. For example, Richardson et al. [8] have shown that the oscillatory phase of our rocking in a rocking chair matches spontaneously with the oscillatory phase of a person seated next to us. We also tend to synchronize the pace of our steps as we walk side by side [9]. The same is true for the respiratory rhythm. Codrons et al. [10] found spontaneous synchronization of body movements and respiratory rhythm between naive members of a group. At the subjective level, Stern [11,12,13] develops the idea that resonance applies also to the temporal contours of our affective experience. The multidimensional studies by Fuchs and colleagues follow this perspective [1,14,15]. From the perspective of neuroaesthetics, the phenomena of mirroring, coupling/synchronization or coordination specify the forms and levels of resonance that are at play in the reception of an artwork. Freedberg and Gallese [3] argue for the role played by the mirror neuron network in art reception. They note the observed activation of the premotor and motor areas in the reception of painting, whether it is figurative and representing human actions, or abstract, such as the works of Kline [16], Pollock or Fontana [17]. In the field of music reception, studies show an entrainment of the spectator’s body movements to the musical rhythm [18,19]. With regard to the reception of dance, a number of studies have detected increased activity in the somatosensory and motor regions of the brain during the observation of dance [20,21,22,23,24,25,26]. The study by Jola and Grosbras [24] is particularly noteworthy as it has demonstrated the importance of conducting studies with live performances. Using Transcranial Magnetic Stimulation (TMS), they found increased cortical motor excitability in non-dancer participants only when viewing a dance piece in a theater. A video of the same dance piece, observed in a lab, did not produce an enhanced motor excitability. Along similar lines, Konvalinka et al. [27] showed heart rate synchronization between spectators and performers during a fire ritual in a rural Spanish village. These studies point to the crucial importance of the co-presence of bodies in the study of interpersonal resonance phenomena and to the importance of running experimentations in an ecological context (set-ups which better approximate real-life situations). Even though the concept of the spectator as a resonator of the art work finds support in the studies relating to the phenomena of attunement during art spectating, two dimensions inherent to the notion of resonance seem to be marginalized in these studies: (1) The temporal extension of the resonance phenomenon and (2) The factors that condition the amplification of resonance. In the case of art reception, we can ask what modulatory, dynamic, or relational phenomena could play a role in amplifying the resonance between the work and the spectator, and to what extent the dynamics of resonance extend beyond the moment of reception. What echoes remain in the spectator’s body-mind (coined by Shaner [28] as a translation of the Buddhist concept Nāmarūpa, the non-separability of the physical and the mental)? To date, neuroaesthetic studies have tended to focus on three phenomena that could be reconsidered as factors related to resonance amplification in dance: expertise in a form of dance [29], preference for a particular movement [20], and the degree of familiarity with the performers [27]. However, these dimensions are properties of the individual spectator and are not inherent or specific to the actual encounter with the work. The reception of a work of art is a relation of reciprocal action that can move us in our sensation, emotion, perception, taste, and in the way we experience ourselves and the world. It varies according to the disposition and the sensibility of the spectator [30] and the characteristics of the work ([31], p. 604), such as its stylistic characteristic [32] or its conceptual dimension [33]. Furthermore, the impact of the above mentioned three different dimensions of resonance was studied only during spectating (and most often during a very short exposure). The impact on the extension of the resonance once the performance concludes was not evaluated. Finally, in most of these studies, a single measure, dimension or modality was investigated (e.g., brain activity or heart rate), yet both the phenomenological analysis of the lived experience and theoretical considerations [11,12,13,34] suggest that resonance phenomena are trans-modal and multi-dimensional. It seems to us that to fully understand these phenomena we must study resonance simultaneously across different modalities, granularities (time scales) and perspectives (namely 1st and 3rd person), while taking into careful consideration the unique characteristics of each artistic work and the context of its reception.

In the study reported in this paper, we were interested in measuring the prolongation of the echo in the spectator’s body-mind beyond the explicit "end" of the work and the spectator-dependent extent of its amplification. We worked with the French choreographer Myriam Gourfink who has been using a contemplative practice for more than 20 years to produce a unique, extremely slow and continuous dance, based on stretched respiration and body-oriented attention [35]. This study was part of a larger project at the crossroads of aesthetics, phenomenology and cognitive science (labodanse project [36]). In one precedent study, we quantified the respiratory synchronization between dancers and spectators during the choreography, which we interpreted in terms of entrainment or resonance [37]. In a second study published in the same paper, with another group of spectators, we observed an effect of Gourfink’s choreography on a duration estimation task. Here we report the results of a follow-up study, where changes post-performance in temporal cognition and physiological rhythms were evaluated concomitantly (with a new group of spectators). Changes in temporal cognition were estimated using a spontaneous tempo task and a novel apparent motion protocol. The specificity of Gourfink’s work (not tested in our previous studies) was evaluated by administering the same tests to a control group that observed a different choreography. Using heart rate and breathing rate measures before and after Gourfink’s choreography, we wanted to evaluate the relationship between the observed changes in temporal cognition and changes in alertness or physiological arousal. This is the first study to look at this relationship in an ecological setting. Cognitive and physiological changes were correlated with subjective reports provided by spectators post-performance. Before turning to the details of our methodology (Section 5) and the results (Section 6), we will first provide relevant background. In Section 2, we describe the artistic work of Myriam Gourfink. We then turn to discussing cognitive findings and phenomenological conception regarding the perception of short duration (nowness, Section 3.1) and the interrelation of temporal cognition, attention and arousal (Section 3.2). This interrelation, particularly prevalent in contemplative practices (Section 3.3), will provide the context for the specific hypotheses of this study (Section 4).

## 2. Slowness, Continuum and Duration: The Choreographies of Myriam Gourfink

### 2.1. Phenomenological Approach to Myriam Gourfink’s Work

The choreographies of Myriam Gourfink combine extreme slowness, a continuum, and duration. The dancers may take up to 8 min to cross 10 cm [38]. Between immobility and movement; this extreme slowness cancels out all narrativity and gives rise to almost imperceptible changes, a continuous parade of tiny variations. Entangled bodies merge, emerge and re-merge again, creating ceaseless shapes and generating visual multi-stability. The extreme slowness of the continuous movement induces, for some spectators, the phenomena of inattentional blindness [39] and blindness to slow changes [40,41]: « The perception of the viewer is constantly challenged by the invisibility of progressive changes intervening in the passage of what the choreographer names a “posture” to another, under the effect of a slow deformation due to micro-movements » ([42], p. 28). Thus, depending on the modulations of the spectator’s attention, it is only afterwards that he realizes that the scene has changed without having perceived the change itself. The choreographer works towards the “elimination of all spectacular processes in favor of a sharpening of the visual” ([43], p. 136). The dancers do not display theatrical expressions, emotions, or mimicry; their eyes are often distant, never addressed to the public: “every dancer is absorbed by her task” ([43], p. 135). The Gourfinkian movement exteriorizes, gives to see and to feel, or materializes, several aspects proper to the meditative experience. Notably, the continuous observation of change is described as follows by Gunaratana in his treatise on meditation: “It’s watching the changing flow of experience. It’s watching things changing. It is to see the birth, the growth, the maturity of all phenomena” ([44], p.212). For the spectator, the Gourfinkian movement induces specifically phenomenological and perceptual activity related to the emergence of phenomena and their appearance, by playing on “the appearance, the background and the form, the Gestalt” ([45], p. 221).

### 2.2. The Making of Slow Movement: Yoga and Meditation or Attention and Respiration

Gourfink’s unique slow movement in continuous flow consists neither in slowing down, nor in reproducing a slower movement. It is rather the consequence of a specific body-mind technique, Energy Yoga [35,38,46,47,48], that she has been practicing for over 20 years. Somatic and contemplative techniques are by now an integral part of contemporary dance training [49,50]. Overall, contemplative practices can be considered as training of attentional regulation as well as of meta-awareness and self-regulation [51,52]. Myriam Gourfink has the distinction of not reserving yoga and meditation to the studio training but bringing them onto the stage. The choreographer recognizes in her lived experience of her dance two elements that slow down the spontaneous pace of the dancer: (1) the respiration—“it is the exploration of respiration which induces the slowdown” ([38], p. 126), and (2) a circulating body-oriented attention—“we do not decide to start from a slow tempo to execute the movements (…), it’s quite the opposite: the slowness is induced by the work on body exploration and by the consciousness of tiny movements” ([38], p. 126). These two elements that induce slowness are practiced by the dancers of the company in the course of the guided training in Energy Yoga. The study by Warzer et al. [53], using a pointing task before and after a training in Energy Yoga, has shown that the training affects not only the speed of controlled goal-directed segment of a voluntary action (the extension of the arm) but also, and especially, the non-intentional or non-controlled segment of that action (the retraction of the arm). The dancers train for at least 2 h before going on stage, and they continue to practice the different elements of the yoga described above during the dance and so remain in a “state” of meditation in movement during the entire choreography [36]. Gourfink’s dancers actively work to cultivate “the awareness of the delicacy of the transition between past action and present action” ([35], p. 28) and this tension brings about “a broadening of awareness” ([35], p. 28). For a phenomenologist, M. Gourfink’s description of the experience of dance during her choreographies strongly recalls the Husserlian schema of the “present” as made of retention and protention, or anticipation [54]. According to Husserl’s schematization, retention refers to what has just happened and protention to what is just going to happen. It is this immediate past and this immediate future that make up the present moment [54,55,56], but it is rare to perceive them consciously [57]. In the above cited description of the choreographer, protention and retention are explicitly perceived by the dancers in a single moment that seems to be “magnified”. Given the characteristics of Gourfink’s work, the reports of spectators, and in light of the phenomenological and psychological literature reviewed below, temporal perception and in particular the perception of short duration (nowness) seemed to us to offer a window into the prolongation of interpersonal resonance after the performance.

## 3. The Shared Present: The Specious Present as a Window into Interpersonal Resonance

### 3.1. The Present Moment

The present time that we perceive (nowness) occupies a certain depth of duration [54,55,58,59,60], otherwise we could not understand a spoken sentence, for example. This phenomenon, involving the fusion of successive elements into a coherent whole, has been referred to as the “specious present” [55], the “psychological present” [61], or the “subjective present” [58]. According to Pöppel [58], this mechanism of temporal integration is automatic and presemantic; it is also operative in movement control and other cognitive activities [58]. Estimations of the width of the specious present are highly variable, from 3 to 8 s by Michon [61], at most 5 s by Fraisse [62]. On average, it is estimated at 3 s [58,59].

In the domain of milliseconds to seconds, that corresponds to nowness, the psychological study of temporal cognition is composed of three main sub-fields, each investigating a specific subjective phenomenon or behavior: the perception of duration, the perception and the production of rhythms, and the perception of temporal order and simultaneity. The studies related to prospective estimation of short durations show variations in the perception of duration that depend on a combination of intrinsic factors, including cognitive, emotional, and body states [63,64], as well as extrinsic factors [65]. For instance, a frightening stimulus is judged to be longer than a neutral or pleasant stimulus [66]. The authors of these studies explain these changes in terms of an adaptive modulation of vigilance and attention processes in response to environmental conditions [67]. In addition, it has been discovered that estimations of duration are modulated by the explicit speed of another’s movement [64,68,69]. However, it is also affected by the implicit rhythm of the other: a stimulus with an image of an elderly person is considered longer than a stimulus of the same duration presenting a young person [70]. Nather et al. [71] found that subjects overestimated the duration of a stimulus presenting a dancer in an arabesque posture to a greater degree than the duration of a stimulus presenting the same dancer standing in a relaxed posture.

Spontaneous tempo (or internal tempo) has been studied within the sub-field of temporal cognition centered on the perception and production of rhythm. Spontaneous tempo refers to an internal personal rhythm. This notion comes from the body of research on the perception and production of rhythm/pulse/beat, particularly in musicology, another field where subjective time has been studied. In that field, a notion of a subjective, or spontaneous, internal rhythm has been elaborated to account for individual differences in rhythm production and perception and more generally to capture certain universal properties of tempo perception and music styles [72]. The existence of an “optimum” in the perception and production of tempo [62] has been used to support the resonating theory of tempo perception. According to this theory, the perception and production of tempo is strongly linked to whole body movements, with the human being functioning as a system resonating with a natural frequency (underlied by a damped resonating oscillator in the perceptual-motor system) [73]. The rhythm of gestures belonging to the repertory of daily repetitive movements, such as walking [74], clapping hands, or tapping with fingertips, is at a frequency of about 2 Hz [75]. Laboratory studies have also found a value of about 2 Hz for both the production and the perception of spontaneous tempo (ST), suggesting that they shared a common internal tempo [72,76]. While the spontaneous motor tempo is considered relatively stable for each individual (varying around 2 Hz for young adults and slowing down with age [76,77]), it presents variations under certain conditions that can be considered as reflections of our “body states”, in continuous interaction with the internal and external environment. As with duration estimation, it has been shown that spontaneous tempo is influenced by external or internal events (external stimulation, or changes in cognitive-emotive state or physiological arousal). For example, a physical exercise such as pedaling increases the heart rate and spontaneous tempo [78]. The spontaneous tempo of walking accelerates or slows down according to an external musical rhythm [79]. In a series of papers, Bove and colleagues have shown that spontaneous tempo can be modulated by the observation of a rhythmic action performed with a fast tempo (3 Hz) or a slow tempo (1 Hz ) [80,81,82]. This modulation was observed even two days after the experimental session. Oullier et al. [83] conducted experiments with dyads performing rhythmical movements. The two participants had to tap at their spontaneous tempo and were not instructed to synchronize. The experiment compared two conditions: blindfolded vs. sighted. They found that spontaneous synchronization arose as soon as visual information was available and was conserved even after vision was occluded again. These findings indicate that motoric patterns may be influenced by another during visual interaction and retain a trace of this impact after the interaction. It has been shown that spontaneous tempo is linked to perceived duration [84], but this relation has not been studied extensively.

The succession or simultaneity of perceived events and the perception of bistable images (e.g., Necker cube, [85]) have been investigated in order to understand the structure and the duration of the temporal interval constituting the subjective present time [58,61]. In this subfield, the apparent motion illusion has been a paradigmatic case study. When two visual stimuli are presented in succession at non overlapping locations, changes in the length of the temporal separation between them will bring about different experiences for the observer. When the separation is very short, the observer will experience simultaneity of the two stimuli. When the separation is long, the observer will experience the two stimuli as following each other, but with intermediate separation, the observer will perceive a single stimulus moving between the two locations. Ichikawa and Masakura [86] have shown that auditory stimulation affects the perception of apparent motion. Nonetheless, we do not know of any studies exploring the impact of interpersonal conditions on the perception of succession or simultaneity.

### 3.2. Modulations of Our Perception in the Present Moment—Attention and Vigilance

Two factors are fundamental to the variations of subjective perception of temporal intervals related to nowness: attention [62,87,88] and alertness (physiological arousal [89,90]). It has been shown that the quantity of attentional resources, the stability of attention, and the direction of attention all impact the perception of the present moment. Dual task paradigms have shown that stimulus durations are judged longer when the amount of attentional resources allocated to time increases and conversely judged shorter when the amount of attentional resources decreases [91,92]. Furthermore, it has been shown that when the amount of attentional resources available for the processing of temporal information increases, time is judged longer and less variable [87,91]. Attentional capacity also plays a role in the formation of the illusion of apparent motion [93,94]. Marusich and Gilden [95], using an apparent motion task, found that adults with Attention Deficit Hyperactivity Disorder (ADHD), one of whose traits is attentional instability, had a shorter temporal integration window compared to normal adults. Puyjarinet et al. [96] found that adults and children with ADHD do not produce significantly different tempi in a spontaneous tempo task compared to a control group. However, the inter-tap variability was significantly larger in the ADHD group, providing additional evidence for the relationship between attention and the size of the temporal integration window. In addition, Pollatos et al. [97] have found that the direction of attention matters (interoceptive vs. exteroceptive), independently of the characteristics of the stimulus presented. When attention is directed to bodily sensations, time distortions are more pronounced.

Physiological arousal or alertness, also called vigilance [98], is often measured as some combination of three physiological rhythms (heart rate, breathing rate, and EEG). Indeed, physical activity, which increases arousal, induces an overestimation of objective duration [99]. This is also the case for emotions like fear [64], whereas certain drugs, such as marijuana, induce an underestimation of objective duration in a reproduction task [100]. Spontaneous tempo increases with stress induced by sounds [101] or by physical activity [102] such as pedaling [78] and decreases through relaxation [101]. However, there is a debate about the direct relationship between changes in physiological rhythms and changes in estimation of durations. Cahoon [90] found a correlation between indicators of arousal and two methods of duration estimation (verbal estimation and production) and one tapping task. However, more recently, Schwarz et al. [103] have found that heart rate changes did not directly impact temporal distortions. We do not know of any studies that have explored the effect of arousal on the perceived illusion of apparent motion. However, it has been shown to be affected by auditory stimulation [86].

### 3.3. Meditation and the Present Moment

Meditative practices can be distinguished from each other by the type of “attentional exercise” that they propose. Two main forms of attention are distinguished in meditative practice: focused attention and open awareness, also known as open-monitoring [51]. Focused attention often favors one of our senses and consists in focusing our attention on a particular object ([104], p. 67). This object may be in the “external” environment, for example a visual focus on the flame of a candle or an image, or the auditory focus on a sound, or it may be “internal”, such as a mantra. Often it is breathing and it’s movement that serve as the locus of one’s focus ([104], p. 62). In mindfulness meditation, or Zen, the attention is global and open (open monitoring), welcoming without judgment internal and external stimuli (sounds, smells, thoughts, emotions…), and their variation from one moment to the next. Another aspect that fundamentally differentiates the different techniques of meditation lies in the control of breathing (or absence there of), in the active (or non-active) participation of the imagination, and in the posture (lying down, sitting, standing, walking, dancing…). For example, Zen technique does not summon the imagination; the meditator focuses strictly on the internal sensations and the sitting posture ([105], p. 199). Other techniques, such as “loving kindness meditation” ([104], p. 84), call for imagination and visualization, getting closer to the techniques of hypnosis or guided imagery [106]. Several recent studies have investigated the effects of meditation on the perception and estimation of short durations. The study by Kramer et al. [107] using a bisection task, showed an overestimation of durations after 20 min of a meditation practice involving a focus of the attention on breathing. In a recent study, Droit-Volet and Heros [66] showed that the immediate effect of the mindfulness meditation session was to reduce the variability of temporal judgment in all participants. Sauer et al. [85] showed that meditators stabilize a bistable image stimulus for a longer period of time, which can be interpreted as a longer duration of subjective nowness. The results of the study by Carter et al. [108] go in the same direction, showing a prolongation for Buddhist monks of the percept of one of the two images in a bistable image. The authors of all these studies explain the lengthening of subjective duration as a consequence of the increase of attentional resources after the practice of meditation. This interpretation is reinforced by the study conducted by Wittmann and Schmidt [109], in which experienced meditators performed time estimation tasks but without having meditated just before. The results do not show a change in estimate of the durations, suggesting that the previously cited results are a consequence of the specific state brought about by the practice itself and not the reflection of individual level traits of practitioners. A number of studies show a slowing down of breathing rate (BR) after a meditation practice. For instance, Lehrer et al. [110] found that respiration rates fell after 10 min of breathing-centered meditation practice (Zen) in experienced meditation practitioners. More generally, meditative states have the peculiarity of presenting both an increase in attention and a reduction in arousal level [111]. We do not know any study on the impact of meditation practice on spontaneous motor tempo. However, in his study about the impact of relaxation, which is a component of the “meditative state”, on tapping, Cahoon [90] found a relationship between breathing rate and tapping. In the low induced arousal condition, the rate of tapping was significantly faster in the high-respiration group than in the low-respiration one, and verbal estimations of durations were significantly lower in the high heart-rate group than in the low one. In sum, the meditative state seems to expand the “specious present” through an increase and a stabilization of the attention, and through a reduced arousal.

## 4. The Shared Present and Interpersonal Resonance

The present or nowness appears as a highly flexible tuning process [61]. Its variations, through modulations of attention and vigilance, reflect our change of body-mind’s states in interaction with the environment and more specifically with others. Temporal perception is fundamental for perception, action [112,113,114,115], and decision-making [116]. Our relations with others (verbal and nonverbal) are organized over time, and this is the case from the first interactions between the baby and her mother [117]. Stern [11,12,13,118] and Trevarthen [119] have argued for the existence of primary intersubjectivity in infants based on rhythmicity of performances (e.g., movements, facial gestures, vocalization). At the crossroads of phenomenology and cognitive sciences, more than being “in” the same moment of objective time (clock-time), intersubjective temporality is understood as the shared experience of time and sharing temporal structure of action and affect [11,12,13,19,117,119,120,121]. In this vein, Tschacher et al. [122] propose intersubjective movement synchrony as a window into shared nowness.

In summary, it seems to us that temporal cognition offers a measure that is both intersubjective, non-verbal, valid in the laboratory and in the ecological context (a non-laboratory setting containing the complexity and richness of real world situations) [75], and reflects our global state in interaction with the environment and with others, making it a relational measure. Moreover, it varies according to two factors that are at the heart of Myriam Gourfink’s choreographies and that underlie their meditative state: attention and physiological rhythm.

### Resonance with Gourfink’s choreographies

Audience reactions to Gourfink’s choreographies vary widely. Some viewers are bored and find time to pass very slowly during the choreography; others, on the contrary, feel that time has passed very quickly. For the latter, aesthetics studies on spectating her choreographies report a number of body-mind effects, including a change in time, space, body perception, and, in particular, an expansion of body boundaries [123,124,125], comparable to those observed in meditation [126]. Gourfink’s choreographies and the state of her dancers are meditation-like, and so, in the framework of resonance, we predict an echo of this meditative state for some spectators after the choreography. In the light of studies on the impact of meditation on the perception of time, we predict that the echo of the meditative state of the dancers and choreography will result, for the spectators, in a slowing down and stabilization of internal tempo, an expansion of nowness and a decrease in arousal in physiological terms. However, as discussed earlier, the experience of a specific art’s work is not universal nor inherent to the work of art but a relationship, so we expect the extent of resonance to vary between individuals. In order to probe this variability, spectators filled in a Gourfink specific questionnaire. The questionnaire was designed to allow us to explore the relationship between eventual cognitive and physiological changes on the one hand and the spectator’s appreciation of the piece and the modulations of his attention during the performance on the other.

## 5. Methods

We administered two measures of temporal cognition before and after a 40-min live performance to an audience (n = 12): a Spontaneous Motor Tempo (SMT) task [76] and a task assessing the temporal window inducing the Apparent Motion effect (AM) [95]. The same temporal tasks were tested with a custom created control choreography with a separate group of subjects (n = 13). All participants provided written informed consent in compliance with the Declaration of Helsinki. Given the ecological context of the testings, it was impossible to perfectly match the demographics of the two groups. The Gourfink group’s mean age was 32.6 (SD 8.2) and was composed of 80% females. 52% of the group had no dance experience with a mean of 6.5 years of dance experience across all participants (SD 7.8). The control group’s mean age was 38.8 (SD 12) and was composed of 70% females. Only 23% of the participants had no dance experience with a mean of 12 years of dance experience across all participants (SD 13.9). The control choreography, created by Namiko Gahier-Ogawa and Deva Macazaga, had the same length and overall structure of Gourfink’s choreography (a solo, followed by a duo, then a second solo and finally a second duo, each of about 7 min long) but was not based on a contemplative practice (no voluntary control of breathing or attention) and contained various rhythmic variations. Both choreographies were presented without music in a dance studio. The spectators were sitting at about 5 m distance from the dancers. Experiential reports concerning attention, temporal modulations and subjective evaluation of the piece were collected post performance to explore the relation between eventual changes in temporal cognition and specific dimensions of the lived experience of spectators. Physiological data (heart and breathing rate) were also collected as indicators of physiological arousal. While data was collected also during the performance, here we analyzed only data from pre- and post- performance intervals. The two groups of spectators followed a 2 h pre-performance movement session. These sessions were built to offer spectators a training close to the one used by the dancers before performance, to provide an embodied insight into the experience of the dancers and the quality of the choreography. The experimental group participated in an Energy Yoga session with Myriam Gourfink. The control group took part in a movement training custom designed by the two choreographers/performers of the dance piece and Sophie Blet, a dance teacher. Analyses of the relationships between physiological rhythms, subjective reports and temporal cognition were performed only for the spectators of Gourfink’s choreography, since physiological measures were not recorded during the control session.

### 5.1. Spontaneous Tempo Production

**Procedure:** We used an adapted version of the McAuley et al. [76] protocol, a standard spontaneous finger tapping task, to evaluate to what extent the observation of Myriam Gourfink’s choreography modulated the SMT of spectators. Participants sat at a table on which the tablets were placed. The experimental session consisted of three blocks: (1) Tap as fast as you can in a regular fashion (for a total of 100 taps), (2) Tap as slow as you can in a regular fashion (for a total of 20 taps), (3) Tap in your internally paced tempo (for a total of 70 taps). The experiment was implemented on tablets using the OpenSesame [127] software. Written instructions were presented on the tablet before each block. The spontaneous tapping task was explained as follows: “each one has their own internal tic-tac. Concentrate on yours and tap on the screen accordingly, tapping on both the tic and the tac”. This instruction was developed by our team after participants in an earlier study found a more terse instruction (tap according to your own rhythm) too opaque. The tablet touchscreen was used to record the Inter Tapping Intervals (ITI). The same task was administered twice: once before and once after the performance.

**Data processing and analysis:** We analyzed data only from the spontaneous tempo block (as in McAuley et al. [76]). Outliers, defined as ITI longer than 3000 ms were removed. SMT varies generally between 500 and 1500 ms so we considered any value larger than 3000, which represented only 0.009 of the data, an outlier due to technical problems with the input device or similar issues. In addition, the data of two subjects in the Gourfink condition was removed since their mean ITI either doubled or halved between the two sessions. We took this to reflect not a change in the underlying SMT but rather as a change in strategy: tapping on both the tic and the tac (strong-weak) or only on the (strong beat) tic, i.e. change in the metrical level. Data from two subjects in the control condition was missing for technical reasons. We performed two different analyses on the data: changes of ITI duration (taken as a proxy of SMT), and changes in individual ITI variability, following the performance in both groups. ITI duration was modeled using a Linear Mixed Model with session (before/after the performance) and group (Gourfink/Control) as fixed effects, as well as their interaction and subject as random effect (lme4 package for R [128]). Individual coefficients of variation (cov: standard deviation divided by the mean) were used to estimate ITI variability before and after the performance and individual differences in variability (pre/post performance) across the two groups were calculated. We used the coefficients of variation as it normalizes for differences in mean. A one-tailed non-parametric Mann-Whitney U test comparing the Gourfink group to the control group was used to test our prediction that variability will be reduced by spectating Gourfink’s choreography. We used a nonparametric variant of the standard t-test because of the relatively small number of subjects in each group.

### 5.2. Temporal Window of the Apparent Motion Illusion

The literature distinguishes two types of apparent motion illusion: short range and long range. Short range apparent motion requires a close proximity between the two stimuli and a very short Inter Stimulus Interval (ISI), lower than 100 ms, and seems to be dependent on low level motion detection mechanisms in the visual system. Long range apparent motion, which we make use of here, involves stimuli in greater distance, longer ISI of up to 500 ms or more, and appears to require attention in order to bind the two stimuli into a continuous movement event [93]. Finlay and von Grünau [129] have systematically explored the size of the spatial and temporal window of the apparent motion illusion involving two flickering dots. They have established that with four degrees of spatial separation, a 500 ms ISI (2 Hz) produced the most stable illusion for normal adults. Given the effect of Myriam Gourfink’s work on (certain) spectators’ experience of nowness discussed above, we hypothesized that after the performance we will observe an increase in the size of the integration window allowing for the emergence of the illusion, corresponding to an expansion of the interval of the present moment.

**Procedure:** Participants were seated at a table, on which the tablets were placed. The OpenSesame software, implemented on 7.85 inch tablets, was used for Stimuli presentation and for recording of the responses. On each trial, the participant viewed a 5 s sequence of alternating white circles (spaced 2.5 cm from each other—over a visual angle of 3°) on a black background. The alternation frequency varied across trials (6 different frequencies: 0.75, 1, 1.25, 1.75, 2.15, 2.75 Hz). Each experimental session consisted of 36 trials, 6 trials per frequency. After viewing each sequence, the participant was asked to indicate his/her subjective response to the question “Do you see two circles that alternate or a circle which moves?” on a continuous scale (from “two circles that alternate” to “a circle which moves”). The task was administered before and after the performance.

**Data processing and analysis:** Subjective ratings were normalized and outliers (1.7%) were removed (values greater/smaller than 2 standard deviations from the mean). Given the small number of subjects, the relatively small number of trials per frequency, and the novelty of the design, we chose to perform two non parametric tests: The first test was carried out in order to evaluate whether the frequency of alternation had an effect on rating. This test was important to validate the design, independently of the effect of performance and group. Average rating per frequency and per subject (collapsing over sessions) were entered into a Kruskal Wallis Test (non parametric ANOVA) with Frequency as factor. The second test was performed in order to evaluate the specific effect of the Gourfink performance compared to the control condition on AM reports. Mean ratings per subject and per session were calculated across all frequencies and a per subject difference measure (post performance mean−pre performance mean) was obtained. A one-tailed non-parametric Mann-Whitney U test was used to test our prediction that the apparent motion illusion will be stronger for the same frequencies of alteration (which we interpret as increase in the size of the integration window allowing for the emergence of the illusion) after the Gourfink performance compared to the Control performance.

### 5.3. Physiological Measurements

Subjects were fitted with Bioharness 3 sensors (Zephyr, USA) during the entire experimental session. We extracted the mean heart rate (HR) and breathing rate (BR) during the 15 min just before and just after the performance (corresponding to the time in which the cognitive experiments were conducted). We did not collect data from the control group and so the reported data only includes measurements from the Gourfink group (n = 11).

### 5.4. Subjective Reports

The experience of any art work is not universal and varies according to each viewer. This is particularly the case in relation to Myriam Gourfink’s choreographies, which do not present any suspense, surprises, spectacular effects, change of rhythms or narration. Specifically, the spectacle of extreme slowness during 28 min provokes very varied reactions: fascination for some, boredom for others. In order to probe this variability, we built a Gourfink specific questionnaire. Post- (Gourfink) performance, subjects rated their adherence to a list of statements concerning their experience during the performance on a 1–5 scale (from “I totally disagree” to “I agree completely”). The questionnaire was constructed according to our research method, at the crossroads of phenomenology, cognitive science, and lived experiences [36]. In order to stay as close as possible to the spectators’ experiences during live renditions of Myriam Gourfink’s dance pieces, first-person data formed the raw material of our questions: both our own experience (in our immersion in the practice of Myriam Gourfink), but also input from spectators in earlier sessions, and from the dancers themselves [130,131], in order to better understand the resonance between their experience and those of the spectators. Finally, we relied on the experience and analysis of dance theorists who studied the reception of spectators of Myriam Gourfink’s choreographies (([42] p. 28), [123], ([43] p. 137)). A semantic analysis of these studies provided us with terms and expressions to formulate more suggestive questions to the participants-spectators. These studies all point out that the spectators, depending on the relation they form with the work, can potentially experience the phenomenological attitude of epoché, which includes paying attention to their own attention. The full list of 18 statements is provided in the Appendix A. Below we present the motivation behind the different questions, grouped into three themes. The questionnaire was constructed to collect the spectator’s experience and evaluation of the choreography (1) but primarily to provide us first person perspective regarding two other dimensions of their experience (related to the amplification or the extension of the spectator’s interpersonal resonance): temporal modulations (2) and attention (3).
**Evaluation of the choreography/dance:**Several studies in the field of neuroaesthetics, have shown a relation between the judgment of taste (I like/I do not like) regarding a dance sequence and the physiological response while spectating it [20]. Given these studies, and given the sometimes radical differences between spectators in the appreciation of the choreographies of Myriam Gourfink, it seemed to us essential to query their personal appreciation, through the following question: “I enjoyed the performance”. Alongside judgment of taste, we included other questions not probing the fine granularity of the spectators’ lived experience during the choreography, but rather their overall appreciation and understanding of the choreography. This section was constructed based on the cognitive interpretation section of the Audience Response Tool (ART) questionnaire ([26]). The ART questionnaire was created in order to evaluate the impact of choreographic intention, knowledge and dance expertise on psychological reactions to contemporary dance. According to the authors, cognitive interpretation involves “the recognition, attribution and understanding of character, qualities and meanings in a particular contemporary dance work”. We have extracted and modified some questions from their questionnaire, for instance “I found these performances intellectually stimulating” and added others such as “this piece tells a story”.**Temporal modulations and qualities: rhythm, duration, speed, flow:**As we discussed in the introduction, intersubjective temporality does not correspond so much to being in the same moment of objective time (clock time) but rather the shared experience of time and sharing temporal structure of action and affect [11,15,19,117,120,132]. We included questions that sought to explore the lived experience of temporality and rhythm for the audience. For example, through the following statements: “The movement of the dancers evoked water and waves”, “The rhythm of my body slowed down”, “I had the impression of going on a trip or a promenade during this performance”. Most of these declarations came from the input of spectators from precedent sessions.**Attention:** The spectator’s attention plays a preponderant role in the aesthetic experience [133] and in the experience of art, all artistic categories combined [134]. Nevertheless, each choreography solicits and directs the viewer’s attention differently [135], and the co-presence of the bodies specific to the live performance invites us to wonder if the singular presence of the dancer underpinned by his attention in the present moment [42] may inform that of the audience [136]. In this section, we looked at three different dimensions of attention: the degree of absorption in the choreography (3a), the directions of attention (3b), and the degree of “attention to attention”, in other terms, of meta-cognitive attention (3c). According to aesthetic theories, the degree of absorption in a work, regardless of the artistic category and whether it is a classical or contemporary work, is one of the first factors of the aesthetic experience and the resonance with the work [31,137]. As M. Massin says in her analysis of aesthetic experience in relation to contemporary art, we do not have such an experience without fully engaging with the work [134]. The two others dimensions are specifically related to the choreographies of Myriam Gourfink and to the meditative state of the dancers. One of the effects of the choreographies of Myriam Gourfink, reported by all the phenomenological studies, is that spectators’ attention travels “from the most intimate corners of their bodies, revealed by the dance, to the performers’ uninterrupted movements” ([125], p. 172). In addition, we know that inward directed attention (internal focus) is increased for the dancer and that they pay attention to their breathing during all the choreography. That is why we wanted to explore the directions of attention of spectators, especially towards their own body (internal focus) and/or towards the body of dancers (external focus). Thirdly, the attention to one’s own attention (meta-cognitive attention or meta-awareness [52]) is a constituent element of the meditative state and has been reported to arise, for some spectators, during the observation of Myriam Gourfink’s choreographies.
(a)**Degree of absorption:**Absorption is a form of attentional modulation, a form of concentration during which the external world is momentarily withdrawn [105], independent of the content of absorption, as in rumination, mind wandering, “flow” [138], hypnosis, meditation, day dreaming or sleep [105]. Absorption is often characterized by “timelessness” and “self-forgetfulness” [139]. For instance, there can be a “forgetfulness” of the “muscular” body and body posture, as is the case when one is so immersed in a fiction, or a video game that one adopts a bad posture without noticing, with the muscular contraction becoming apparent only once out of absorption. In relation to the choreographies of Myriam Gourfink, in which the dancers are in a form of absorption, some spectators report being absorbed in the choreography while others do not manage to “enter the dance”. That is why we have tried to interrogate the degree of absorption of the spectators, but also the quality and modality of the absorption, given that attention can be decoupled from perceptual external input [140]. Items concerning absorption included: “I would be incapable to say if I was sitting comfortably or not”, “During the performance I was preoccupied with personal problems”, “I fell asleep during the performance”.(b)**Direction and object of attention:** Joint attention to an object, a stimulus, or a portion of the external environment is related to interpersonal resonance processes, and is a key step in the development of "social cognition" in children [141]. Regarding the attention to others, multiple studies have explored the impact of visual attention directed towards the other (their action [142], their face, their eyes [143], or their mouth [144]). Some recent studies have focused on mutual attention through mutual gaze and have shown that it impacts the neural synchronization between participants [6]. No study has investigated the effects of an individual’s internally oriented attention on the attention of an outside observer. However, Myriam Gourfink’s choreographies are based on extremely precise scores for the dancers’ outer movement but also on the circulation of their internal attention. The attention of the dancers is essentially directed towards several points of their body, as well as to their own breathing during the entire choreography. In this section, we wanted to explore the entrainment of the spectator’s attention by the attentive disposition of the dancer to bodily sensations. In particular, we wanted to explore whether the dancer’s breathing is an object of joint attention (“I was often attentive to the breathing of the dancers”), and/or, if there is an entrainment of the viewer’s attention by that of the dancer (“Many time I felt my own breathing”), and if so, how it relates to changes after spectating. Indeed, in our previous study we found that the strength of breathing entrainment between dancers and spectators during the choreography positively correlated with the responses post spectating regarding spectators’ attention both to the dancers’ breathing and their own breathing [37].(c)**Extent of meta-cognitive attention:** As we have described above, spectating Myriam Gourfink’s work can increase the spectator’s meta-cognitive attention such as paying attention to one’s own attention, which is a feature of the phenomenological attitude (epoché) and contemplative practices. For instance, all spectators experience moments of blindness to progressive and slow changes while spectating Myriam Gourfink’s choreography, but not all of them notice it. The questionnaire included items that explored the extent of spectators’ meta-cognitive attention during the performance such as: “Sometimes the movement was so slow that I did not see the change from one posture to another”.

## 6. Results

### 6.1. Cognitive Tasks

#### 6.1.1. Spontaneous Tempo

Regarding ITI duration (Figure 1A), we observed a main positive effect of the performance. This means a longer ITI after the performance (effect size = 197.12, t = 10.086, *p* < 0.001). There was no overall difference in ITI across the two groups, but as it is clearly seen in the plot and confirmed by the analysis, there was a significant interaction between session and group (effect size = −209.09, t = −8.616, *p* < 0.001). The difference in ITI duration was significantly larger after the Gourfink performance compared to the control performance. The means of the difference in ITI variability before and after the Gourfink and the control performances are plotted in Figure 1B. The performance had an opposite effect on ITI variability in the two groups. ITI variability decreased after performance for the Gourfink spectators, while it increased for the control group. The difference between the two groups was significant (W = 24, *p*-value = 0.0281).

#### 6.1.2. Temporal Window of the Apparent Motion Illusion

As expected, frequency of alternation had a significant effect on AM reports, with stronger AM percept with increasing frequency (Kruskal-Wallis chi-squared = 35.343, df = 5, *p* < 0.0001, Figure 2A). This sensitivity of subjective reports to the frequency of alternation validates our novel design as a measure of the size of the AM temporal window. In accordance with our hypothesis, the difference between AM perception after, compared to before the performance (Figure 2B) was significantly larger for the Gourfink group compared to the control group (W = 21, *p*-value = 0.0079). Though we did not test directly the quantity of temporal expansion (this would have required a protocol with many more trials with a finer grained distribution of frequencies, so too long for our ecological setting), when we look at the mean rating from the Gourfink group by frequency (Figure 3), we observe that the mean rating arrives at the maximum value only with a 2.75 Hz presentation rate before the performance but already at 2.15 Hz after the performance. This shift suggests an expansion of the integration window of about 140 ms. The exact quantification of the expansion will require further testing. In the same figure we also plot the data from the control condition, which do not show an analogous difference between pre- and post- performance.

#### 6.1.3. Comparing the Two Tasks

We correlated the effect size across the two tasks in order to examine whether the changes, observed in both tasks are the consequence of a change in the same underlying cognitive dynamics. For each subject in the Gourfink group, we calculated the difference in ITI (post − pre) and ITI variability (coefficient of variance post − coefficient of variance pre) in the Spontaneous Tapping task. For the Apparent Motion experiment, we calculated per subject the difference (post − pre) in the mean AM rating across all frequencies. Due to the small number of subjects, the correlations are not significant (for ITI: −0.54, *p* = 0.1344, for ITI variability: Pearson r = −0.496, *p* = 0.1749) but plotting the data suggests that these might be meaningful (Figure 4). The numerical negative correlation between ITI variability differences and differences in AM is congruent with the observed differences between Gourfink and control. The more a person’s perception of AM increases post performance, the larger the decrease in their ITI variability. However, the numerical negative correlation between ITI differences and AM differences is surprising: the more a person’s perception of AM increases post performance, the smaller is the post-performance increase in the length of their ITI. Given the small number of subjects and the non-significance of the results we leave this puzzle for further research.

### 6.2. Changes in Physiological Rhythms and Their Relations to Changes in Temporal Cognition

We first computed a linear mixed model to compute the effect of the performance on HR and BR. We then calculated an individual, normalized difference (pre-post) for these two measures, as well as for their coefficient of variance. The breathing rate was significantly slower after the performance (effect size = −1.7476, t = −4.085, *p* = 0.0022, Figure 5A), and its variability significantly higher (effect size = 0.073936, t = 2.812, *p* = 0.01840, Figure 5B). Neither HR nor its variability changed significantly after the performance. In order to investigate the relationship between changes in physiological rhythms and changes in temporal cognition, we correlated the changes in task effect size (ITI, ITI variability and AM percept) with changes in breathing rate, heart rate and their variability (cov, coefficients of variation). The only significant correlation was observed between breathing rate and AM perception. The more a participant’s breathing slowed down after the performance, the more their apparent motion percept increased ( Pearson r = 0.804, *p* = 0.016, n = 9, Figure 5C).

### 6.3. Subjective Reports and Changes in Temporal Cognition and Physiological Rhythms

Individual effect size for AM, ITI and ITI variability, BR and BR variability were computed as above. Because of technical issues, we only had questionnaire responses to a subpart of the subjects. We correlated the individual responses (Pearson implemented in R) with the individual effect size for the three cognitive measures and the physiological measures. Given the small number of participants and the multiple comparisons (we correlated the individual effect sizes with all 18 questions), the p values of the individual correlations are not meaningful and the results and their discussion below should be considered as exploratory. The meaningful correlations between cognitive and physiological changes and assertions relating to the evaluation of the choreography are plotted in Figure 6, with assertions relating to temporal modulations and qualities in Figure 7 and with the different dimensions of attentions in Figure 8, Figure 9 and Figure 10.

## 7. Discussion

The cognitive tests before and after spectating the choreographies of Myriam Gourfink revealed a significant difference between the group of Gourfink’s spectators and the control group with respect to the extent of change in both spontaneous tempo and the degree of reported apparent motion (AM) illusion post performance. This difference argues that these effects were due to the specific quality of Gourfink’s choreography. Regarding spontaneous motor tempo, our results show longer ITI after the Gourfink performance. We interpret this result as a slowing down of the spontaneous motor tempo of the spectators after the performance, a lasting effect due to an entrainment by the choreography. It has been shown that spontaneous motor tempo can be entrained by an external rhythm (of an artificial source or of a partner) and that this modulation can last after the interaction has ceased [82,83]. However, in all these studies, participants were entrained by an explicit rhythmic pattern. In the case of Gourfink’s choreography, the dance movement is non rhythmical and so it is not ob§vious if the results mentioned above can explain the slowing down of spontaneous tempo observed here. An alternative explanation for this effect could be in terms of resonance with the meditative state of the dancers. Though the effect of meditation on spontaneous tempo has never been studied, Cahoon [90] found that a short relaxation practice (one dimension of meditation) brought about a slowing down of spontaneous tempo. If the spectators share the meditative state of the dancers, the slowing down of their spontaneous tempo after the performance could be understood as an echo of the “shared present” with the dancers. Further research is required in order to disentangle these different explanations of the observed results.

In addition to the slowing down of tempo, we observed a reduction in ITI variability after the Gourfink performance and an increase in ITI variability after the control performance. There is little literature on ITI variability in a spontaneous tempo task. Puyjarinet et al. [96] found that, while their spontaneous tempo was not different from controls, children and adults with ADHD exhibited more variable ITI. They explain this increase in variability as a motor control deficit. Relatedly, Naranjo and Schmidt [145] showed that Mindfulness training affects motor control. Training resulted in a significant improvement in motor control during perceptual-motor conflict task. In line with our discussion above, we suggest that the decrease in ITI variability after spectating Gourfink’s choreography is due to the sharing of the dancer’s meditative state. The observed (numerical) correlation between decrease in ITI variability and increase in AM percept provides further support for this interpretation. This relation between the size of the temporal integration window and motor control is not only found in meditation but has been also observed in pathological cases such as Parkinson’s disease, schizophrenia, attention deficit hyperactivity disorder and autism [146]. This connection between temporal integration and motor control resonates with N. Depraz’s neurophenomenological view of attention, which she writes “is not limited to one form of concentrated mental activity, but refers to a more complex, differentiated experience, rooted in bodily attitudes: visual, tactile, kinesthetic and verbal.” ([147], p. 345).

The AM illusion has been claimed to index the length of the temporal integration window that corresponds to the “present moment” [58]. Since participants in the Gourfink performance reported significantly stronger illusion after the performance for the same objective inter-stimulus interval (with an estimated shift of about 150 ms in the crossing of 0), we explain our results as an expansion of the present moment [59]. We propose that the observed expansion of the present moment for the spectators is an echo of the meditative state of the dancers and an effect of the specific quality of the movement during the choreography. So, we consider that it can be understood as a lasting effect, or an echo of the interpersonal resonance. It is the first study where the apparent motion illusion has been used as a measure of change in the temporal integration window as the echo of interpersonal resonance and the “shared present” between dancers and spectators. Following Horowitz and Treisman [93], we attribute the change in the integration window after the choreography of Myriam Gourfink to an increase in the attentional resources and attentional stability of the spectators. Both are characteristic traits of meditative states.

Breathing rate (BR) was significantly slower after the performance, and its variability significantly higher. Neither HR nor its variability changed significantly after the performance. This decrease in BR post-performance might be related to resonance with the dancer’s breathing (see Bachrach et al. [37]), but it might be not specific to Myriam Gourfink’s work. Since we did not record physiological data in the control condition, the specificity of this change cannot be evaluated. It is possible that the mere fact of sitting down for an hour brings about a decrease in arousal. However, the observed, statistically significant, relation between breathing rate changes and change in AM reports (the more a participant’s breathing slowed down after the performance, the more their apparent motion percept increased) suggests a specific impact of Myriam Gourfink’s choreography on breathing rate. Furthermore, changes in arousal have been shown to impact the temporal integration window [63]. In view of this known relation between meditation, arousal and attention, discussed in the introduction, the observed correlation indicates that certain spectators experience a meditative like experience during the performance. We propose that this state is an echo, in the spectator, of the meditative state of the dancers. It is worth noting that this preliminary study is, to our knowledge, the first to investigate the relation between breathing rate and the modulation of the perception of apparent motion illusion.

Breathing rate variability (BRV) was overall higher after the choreography. Our measure of breathing rate variability (the variability of the mean BR per minute during a window of 15 min) is not equivalent to the measure of respiratory variability based on inter-peak data (RV, [148]), even though it is related to it (increase in BR variability entails increase in RV but not vice-versa). There is a very limited literature on the cognitive correlates of RV. Sustained attention and cognitive load, as well as anxiety, have been shown to be associated with a decrease in variability [149,150], while certain emotional states have been associated with increase in variability [151]. Given the paucity of literature, the fact that our measure of BRV is not as sensitive as RV, and that we cannot assess the specificity of this effect to spectating Gourfink’s choreographies, it would be premature to interpret this result on its own. However, we will come back to this finding in the discussion of subjective reports on attention.

### 7.1. The Relationship Between Subjective Experience and Observed Cognitive and Physiological Changes

Given the small number of participants for which we had both subjective reports and quantitative data, this is only a very preliminary exploratory discussion with the purpose of opening future avenues of research. We now turn to discuss the different sections of the questionnaire.

#### 7.1.1. Evaluation of the Choreography

Adherence to the assertion “I enjoyed the performance” was correlated with slowing down of breathing rate following the performance. This correlation between appreciation and change of the respiratory rhythm of the spectators goes in the direction of studies in neuroaesthetics which emphasize the direct relationship between the appreciation of a work and the echo in the viewer at different scales: neural, physiological, sensorimotor and behavioral. This relationship has been shown in music [152,153,154] and dance [20,155]. In our previous study [37], we found that an increase in respiratory synchronization between Gourfink’s dancers and spectators was associated with a stronger adherence to the assertion “I enjoyed the performance”. The current study is the first study to correlate multi-dimensional changes in the spectator post-performance with appreciation. The evaluation section was constructed based on the cognitive interpretation section of the ART questionnaire ([26]). However, Myriam Gourfink’s choreographies do not intend to present a fiction or a narrative piece, in which costumed performers dance in a manner appropriate to a character they embody, or communicate emotions through body movement or face mimicry. In light of this observation, it is interesting to note that an increase in adherence to the assertions “The dancers’ faces expressed a lot of emotion” was associated with an increase in ITI variability. Importantly, it is a *decrease* in ITI variability that reflects a specific echo of the meditative dimension of the choreography. We note then, that subjects that do engage in cognitive interpretation display less resonance with the choreography.

#### 7.1.2. Temporal Modulations

The adherence to the assertion “The rhythm of my body slowed down” correlated positively with an increase in ITI, in other words, with the slowing down of the spontaneous tempo post-performance. This correlation thus suggests a meaningful relationship between the spectators’ “introspection” about their felt body and the slowing down of their internal tempo (spontaneous motor tempo, evaluated through a 3rd person methodology).

#### 7.1.3. Attention

**i. Degree of absorption:** The reception of a work of art is a relation of reciprocal action; it varies according to the disposition of the spectator and the sensitive characteristics of the work ([31], p. 604). Our results suggest that the degree of reported absorption in Myriam Gourfink’s work is indeed associated with post-performances indices of interpersonal resonance. For example, stronger adherence to the assertion “I would be incapable to say if I was sitting comfortably or not” correlated with, at the cognitive level, the stabilization of the spontaneous tempo (decrease in ITI variability), and, at the physiological level, the increase in breathing rate variability. As we have seen above, decrease in ITI variability can be understood as an echo of the specificity of Gourfink’s movement. On the other hand, a stronger adherence to the assertion: “During the performance I was preoccupied with personal problems” was correlated with a lower level of cognitive echoes, both with regard to the perception of apparent movement (decrease in the report of AM percept) and with respect to the stabilization of the spontaneous tempo (increase in ITI variability). This observed relation is reminiscent of the comment made by Dufrenne about the aesthetic experience: “The aesthetic feeling is profound because it gathers us together, it is also because it opens us, because the interior life does not lead the subject through the misty meanders of the subjective rumination” ([31], p. 502).

**ii. Direction and object of attention** We found that stronger adherence to the assertion “Many times I felt my own breathing” was correlated with both cognitive and physiological levels of resonance: the stabilization of spontaneous tempo (decrease in ITI variability), and increase of breathing rate variability. We did not find any correlation with “I was often attentive to the breathing of the dancers” (contrary to our expectations based on our previous study [37], where both internally and externally oriented attention were linked to the increase of breathing synchronization between dancers and spectators). This result suggests that it is the attention to their own breathing that plays a role in the extension of resonance after the performance. Since attention to one own’s breathing is often the first step in any contemplative practice, this correlation between first-person reports and cognitive and physiological indices of meditative state provides further support for the assertion that viewers experience something similar to that experienced in contemplative practices. As breathing is also a continuous focus of the dancers’ attention during the entire choreography, these results also invite us to think of an attentional level of resonance within the larger framework of social cognition and interpersonal resonances ([136], pp. 149–154, [147], pp. 437–486). To date, most studies that explore the role of attention in interpersonal resonance processes have focused on attention directed towards the other. Our results invite future research to look at the impact of attention to oneself, and in particular to one’s breathing, on the relationship to the other. One particular question to be addressed is the role of the slow quality of movement in this inwardly directed attention. Deinzer et al. [156] evaluated the sense of time, and the direction of attention (using the same questions as the one of our precedent study after spectating) while spectating a live dance, with two dance solos that differed in speed of movement. They found that participants liked the fast dance more compared to the slow dance, and that during the fast dance, participants focused more on the dancer’s breathing and less on their own body. However, the spectating of the slow dance increased internally oriented attention.

**iii. extent of meta-cognitive attention (during the performance):** In this section, we wanted to explore the extent of spectators’ meta-cognitive attention during the performance. The adherence to the assertion “Sometimes the movement was so slow I did not notice changes in dancer posture” was correlated with cognitive resonance specific to Myriam Gourfink’s work: stabilization of spontaneous tempo (decrease in ITI variability) and increase in the report of AM percept . This relationship between meta-cognitive awareness (attention to one’s own perceptual experience, as in meditative experience or the phenomenological epoché), stabilization of spontaneous tempo and increase in AM percept provides further support to our interpretation of ITI variability as a marker of one’s attentional state.

### 7.2. Limits of the Current Study

It is important to recognize a number of limitations to our study, both in order to help in the interpretation and appreciation of our results, and especially as issues to be considered by future ecological studies interested in live performance. The first issue is the sample size. Unlike laboratory studies, ecological studies of spectating cannot be (easily) administered to individual subjects, but to groups of spectators. While by using tablet-based technology we were able to test simultaneously a number of subjects, this constraint limits the number of subjects per performance. Performances are live events. Even with a ’fixed’ choreography, the movement itself, and more importantly the lived experience of the dancers and spectators, is never identical across repetitions. As a consequence it is not straightforward to generalize across different presentations of the ’same’ dance (see [157] for a recent study looking at dance spectating where the different ’repetitions’ of the same performance produced very different results regarding spectators’ evaluation and engagement and related measures). While the number of subjects in this study was comparable to more conventional studies of spontaneous tempo and tapping more generally (and allowed us to identify significant effects using standard parametric tests), it was too small to reliably identify correlations across studies or between subjective reports and cognitive changes. A second limitation of our protocol was time. The multiplicity of tasks, the time required for logistical purposes (e.g., placing of sensors) and the duration of the dance performance put a strong constraint on the length of each individual task. This was particularly an issue for the apparent motion protocol. In order to do a full psychophysical evaluation of the impact of frequency on the illusion, we would have needed a much longer protocol that was impossible to implement in the context of our study. A third important issue to consider is the collection of first person data in the context of durational events (in contrast with short stimulations used in most laboratory experiments). Our multilevel study suggests an important relationship between attention to oneself and the other in the amplification and the prolongation of interpersonal resonances. However, the exploration of first-person data remains limited, since the data was collected after the reception, in the form of semi-directed and very global questions regarding the modulations of attention during the choreography. In addition, we have not explored the attentional dynamics experienced by the dancers, limiting our understanding of the extent of attentional resonance between dancers and spectators. In following studies in our group we have been developing on-line data collection methods using tablets and smart-watches (in order to collect real time data from both spectators and dancers) as well as post performance video-confrontation procedures. While technologically these procedures are now easy to implement, even in large scales, the impact of a secondary, meta-cognitive task while spectating or dancing is not trivial and needs to be considered carefully.

## 8. Conclusions

This study investigated the echo of interpersonal resonance processes after spectating a live choreography, in an ecological artistic context, using changes in temporal perception, physiological rhythms and indices of lived experience as probes. Working with the French choreographer Myriam Gourfink, who has developed a very specific dance vocabulary over the last 20 years, based on a contemplative practice, we observed, post-performance, an echo in the spectators of the resonance with the choreography at the physiological, cognitive, and attentional levels. As we hypothesized, spectators of Gourfink’s choreography exhibited a number of observable changes, often associated with a state of meditation or deep relaxation, notably a slowing of the spontaneous tempo, enhanced motor control, widening of the temporal integration window or nowness, and associated slowing down of the breathing rate. We have shown that at least the cognitive changes were specific to these spectators and not observed in spectators of a control choreography. We propose to interpret this apparent meditative state of the spectators as a prolongation, post-performance, of their resonance with the choreographic work. While this is the first study to explore meditative states in spectators of dance or performing arts more generally, Pelowski [158] has studied (using self reports) the impact of viewing Mark Rothko’s paintings on the state of the spectator after the experience. They proposed to frame the experience in terms of the Zen Buddhist concept of Satori (realization of emptiness), and articulate a model that shares many features with the perspective of our study and in particular the role of meta-awareness brought about by the interaction with the work of art. We found the degree of reported meta-awareness during the performance to be related to the extension of resonance post performance. The relationship between interpersonal resonance and meta-attention is not specific to the context of art spectating. In Depraz neurophenomenological analysis, “the attention to attention” is a fundamental dimension of shared attention and intersubjective ethics ([147], pp. 437–486).

While it has been previously shown that the rhythm of another’s movement can impact one’s own internal tempo, an analogous effect on the variability of spontaneous tapping has not been reported, nor a relationship between social interaction and the temporal integration window. Future studies in the domain of contemplative practices, neuroaesthetics and social interaction more generally could benefit from the inclusion of such measures. The correlation, albeit non significative, between increase in AM perception and the decrease in ITI variability suggests a shared process. We suggest that this shared process is the stabilization of attention post performance. The fact that an increase in ITI length is negatively correlated (again, not significantly) with an increase in the temporal integration window suggests that the increase in ITI length and size of temporal integration window after the Gourfink performance are not caused by the same underlying change. Further research is required to elucidate the relationship between these two phenomena and to better understand the underlying mechanisms. Finally, we also observed a statistically significant relation between decrease in breathing rate and increase in the temporal integration window, a relationship that has not been previously studied. We speculate that it is an underlying change in arousal that is responsible for both effects, but further research, both in the lab and in the field, is required in order to further validate this relationship and better understand its cause. Taken all together, our results echo the psychological approach to duration offered by the philosopher H. Bergson [159]. Bergson made a direct link between lived experiences of duration, bodily awareness, body-mind states, and the speed of internal rhythm. According to him, our present reflects “above all the state of our body” ([159], p. 557). The elasticity of our lived experiences of duration is a reflection of the modulations of our body states. He proposed that the speed of internal rhythms could measure “the degree of tension or relaxation of different kinds of consciousness” ([159], p. 275).

Even if the correlations with subjective reports of the spectator’s experience is based on a small number of participants, it seems to us that they offer interesting ways of clarifying the relationships between changes in physiological rhythms, changes in temporal perception and modulations of attention, in the framework of interpersonal resonance. The observed correlations between subjective reports concerning the spectator’s experience during the performance and quantitative changes observed after the performance allow us to weave a preliminary link between the moment of co-presence of the bodies during the choreography and the echo in the spectator after the choreography. This preliminary link offers the possibility to approach two dimensions inherent to the notion of "resonance" that seem to us often marginalized: the temporal extension of the resonance phenomenon and the person-specific factors that relate to it. First, they extend an observation, made in many studies, regarding the important role of the appreciation of a work in the amplification of the resonance with it. Here we have shown that appreciation also plays an important role in prolonging resonance. Our results point to the important role of the modulations of subjective attention in the prolongation of interpersonal resonance. Across subjects, the extent of resonance with the choreography was associated with the degree of absorption, of attention directed body-internally, and of meta-cognitive attention of the spectator. This observation dialogues with many theories in philosophy and aesthetics, which consider that the modulations of attention play a major role in the “aesthetic experience” [31,133,134,136,137]. In this sense, J. Rancière, who analyzes the potentially emancipatory dimension of the experience of art [160,161], notes: “There is no form of emancipation per-se in the existence of a painting, in looking at a painting, there is emancipation (…) in the manner of modulating our attention.” [162].

The experience of an art work is highly subjective and varies according to the context and the individual. In this paper we describe a novel multilevel protocol for the study of dance spectating in an ecologically valid setting. It is the first study of a live performance spectating to combine physiological measures, pre and post testing of cognitive indices and subjective reports. An important dimension missing from the current study, but that we have implemented in subsequent protocols, is the collection of online subjective reports during the performance. Our study highlights the importance, in the field of neuroaesthetics and performing arts research, of conducting experiments in an ecological context. The use of video should be reconsidered, especially when the performance was conceived by the artist in terms of co-presence of the bodies of dancers and spectators. Furthermore, our study points to the importance of conducting experiments of long duration. Indeed, in dance and performance arts more generally, duration and temporal structure are fundamental dimensions. Myriam Gourfink’s pieces last between 20 min and 6 h, never a few seconds as is the case of many experiments on dance conducted in the lab. In the case of Myriam Gourfink, it often takes about 10 minutes to enter the choreography. In this sense, the dancer and philosopher G. Fontaine notes that the expansion of nowness regarding spectating Gourfink’s choreographies is due to a progressive experience: “The dancer’s very full time meets this time of attention/waiting of the spectator who experiences a stretching of time, where each sign takes on an increasingly important value. (…) The spectator evolves little by little towards an extreme mental and physical availability. He is confronted with his own position as a spectator, and his own use of time. If he does not make himself perceptively available to every detail, he misses the dance. But he can also ask himself the question of how far he will make himself available.” ([43], p. 137).

## Figures and Tables

**Figure 1 behavsci-08-00024-f001:**
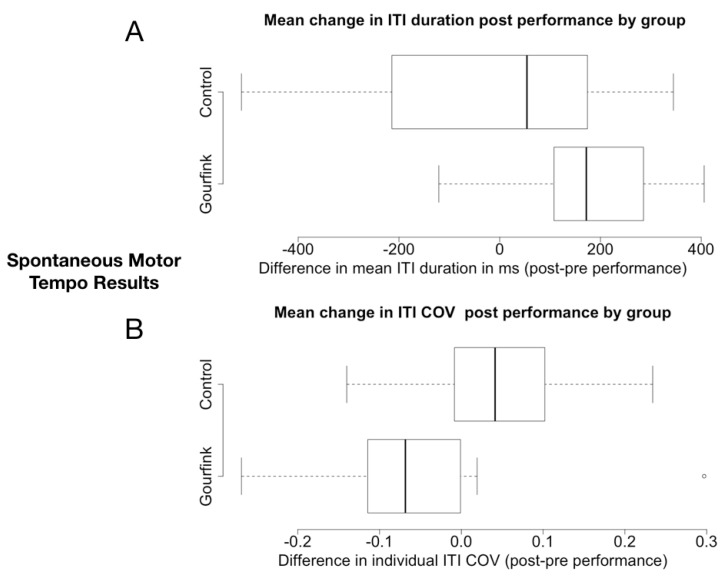
(**A**) Mean change in Inter Tapping Intervals (ITI) duration post performance by group; (**B**) Mean change in individual ITI cov (coefficients of variation) post performance by group.

**Figure 2 behavsci-08-00024-f002:**
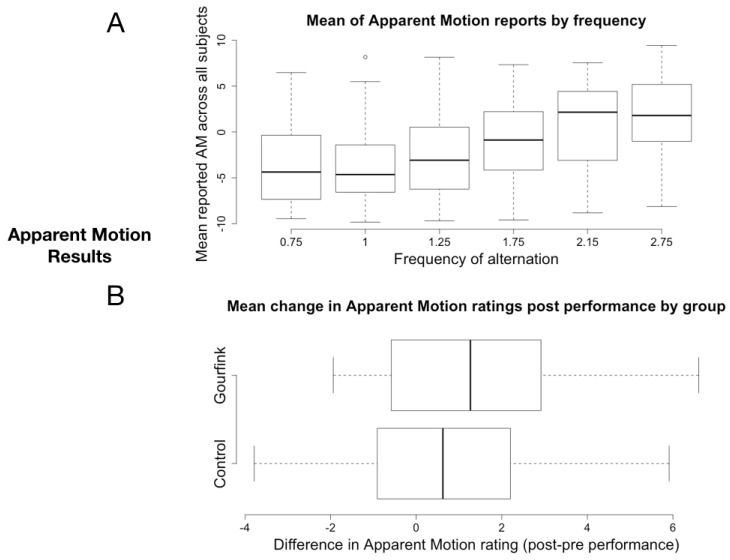
(**A**) Mean Apparent Motion reports by frequency across all subjects and sessions. On the x-axis, the 6 different alternation frequencies used in the experiment. On the y-axis, the mean rating of apparent motion illusion (from −10 to 100 across all subjects and all sessions. (**B**) Mean change in Apparent Motion ratings post performance by group. On the x-axis the difference between the group mean rating (across all frequencies) of apparent motion before and after the performance. Values greater than 0 indicate that after the performance there was a stronger AM illusion.

**Figure 3 behavsci-08-00024-f003:**
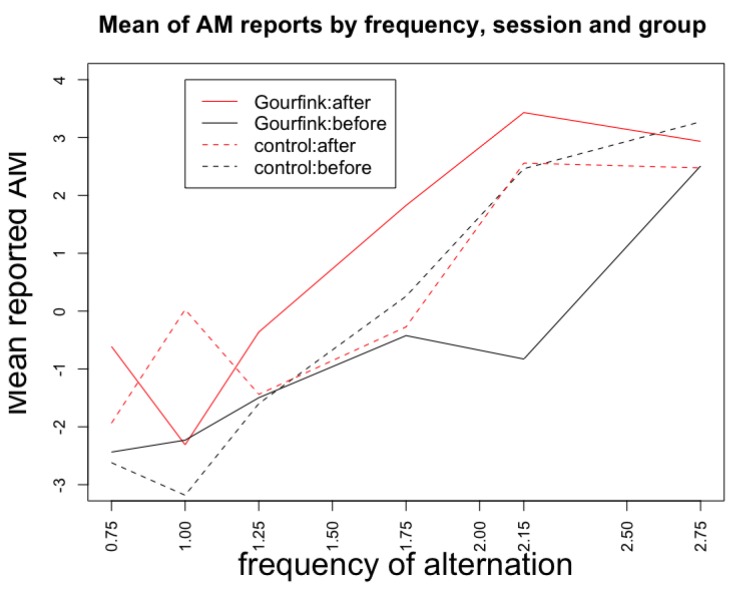
Mean Apparent Motion reports for the Gourfink and control groups by frequency and session. The data were standardized by removing the mean for each group to allow for visual comparison.

**Figure 4 behavsci-08-00024-f004:**
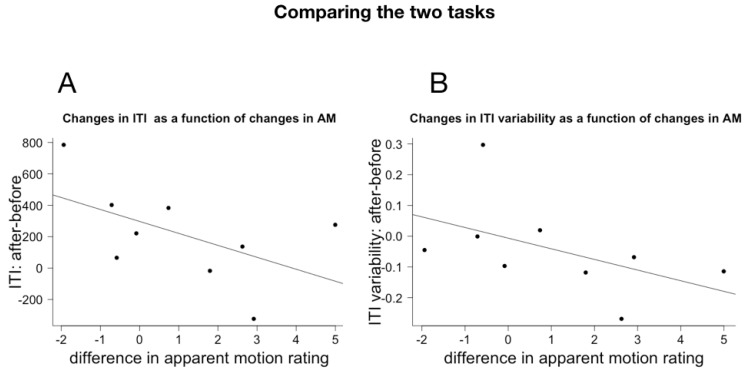
(**A**) Individual changes in ITI as a function of change in Apparent Motion (AM) percept (**B**) individual changes in ITI variability as a function of change in AM percept.

**Figure 5 behavsci-08-00024-f005:**
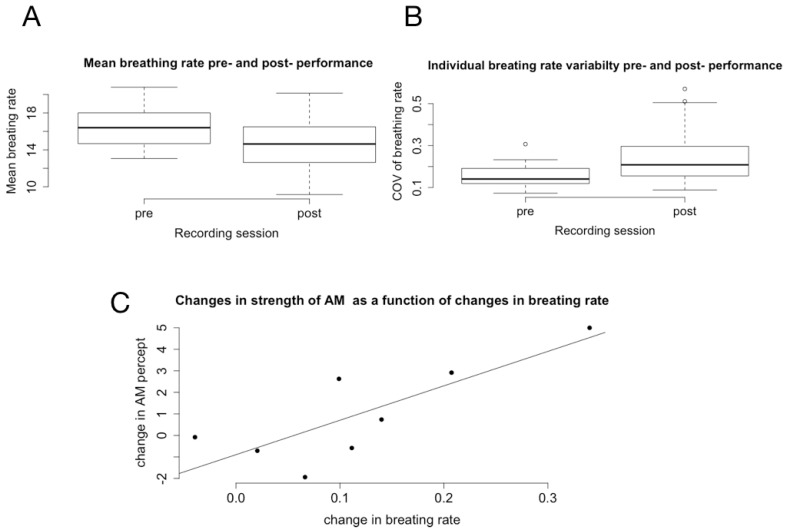
(**A**) Group mean breathing rate before and after the performance; (**B**) Group mean breathing rate variability before and after the performance; (**C**) Change in apparent motion percept as a function of change in breathing rate. On the x-axis change in breathing rate. Values higher than zero indicate a slowing of the breath post-performance. On the y-axis the extent of change (per subject) in the AM report. Values greater than 0 indicate a stronger AM illusion post performance.

**Figure 6 behavsci-08-00024-f006:**
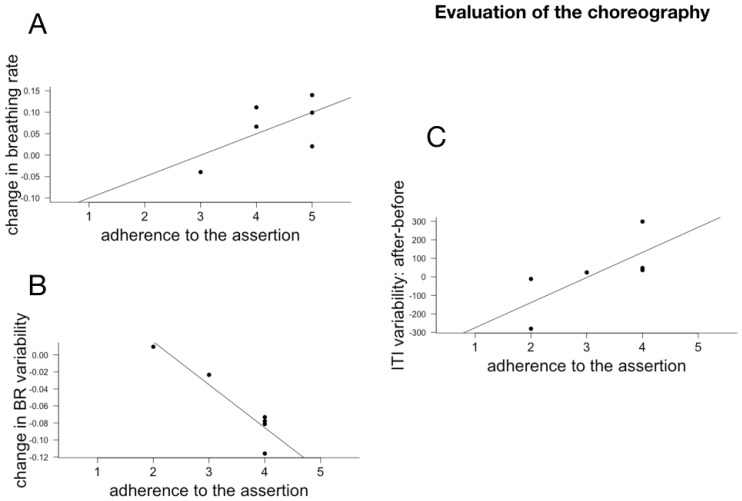
(**A**) Stronger adherence to the assertion “I enjoyed the performance” was correlated with slowing down of breathing rate (0.617); (**B**) Adherence to the assertion “This piece tells a story” correlated with decrease with breathing rate variability (−0.933); (**C**) *Decrease* in adherence to the assertion “The dancers’ faces expressed a lot of emotion” was associated with a *decrease* in ITI variability (0.723).

**Figure 7 behavsci-08-00024-f007:**
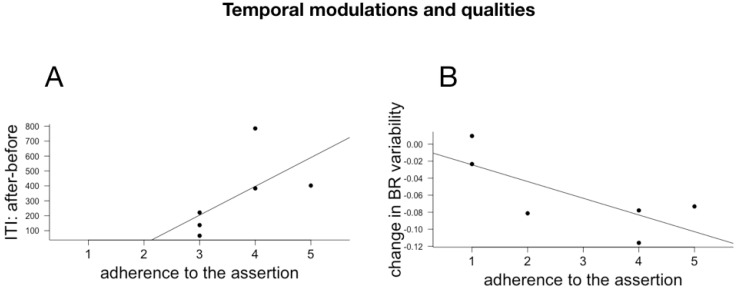
(**A**) The adherence to the assertion “The pace of my body slowed down” correlated positively with Increase in ITI (0.6); (**B**)The adherence to the assertion “The movement of the dancers evoked fluids, water and waves” correlated with increase in breathing rate variability (−0.745).

**Figure 8 behavsci-08-00024-f008:**
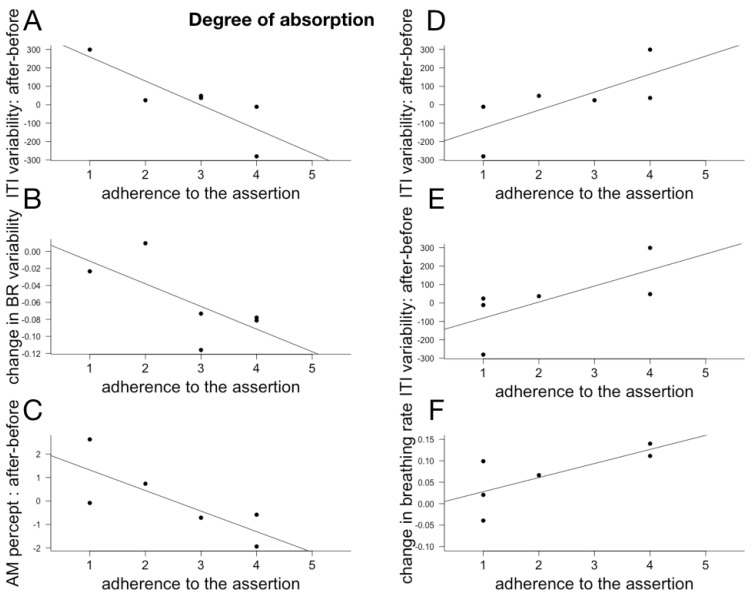
Stronger adherence to the assertion “I would be incapable to say if I was well seated or not during the performance” correlated with decrease in ITI variability ((**A**), −0.902) and with increase in breathing rate variability ((**B**), −0.688). A stronger adherence to the assertion: During the performance I was preoccupied with personal problems was correlated with decrease in the report of AM percept post performance (−0.63, (**C**)) and with increase in ITI variability (0.73, (**D**)). Increase in adherence to the assertion “I fell asleep during the performance” was correlated with increase in ITI variability (0.694, (**E**)) and with slowing down of breathing rate ((**F**),0.732).

**Figure 9 behavsci-08-00024-f009:**
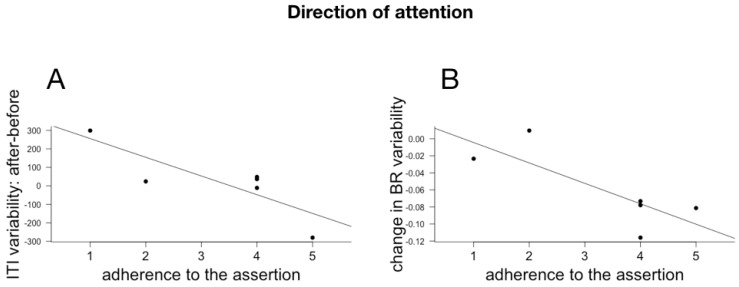
Stronger adherence to the assertion “Many times I felt my own breathing during the performance” was correlated with decrease in ITI variability (−0.878, (**A**)) increase of breathing rate variability (−0.796, (**B**)).

**Figure 10 behavsci-08-00024-f010:**
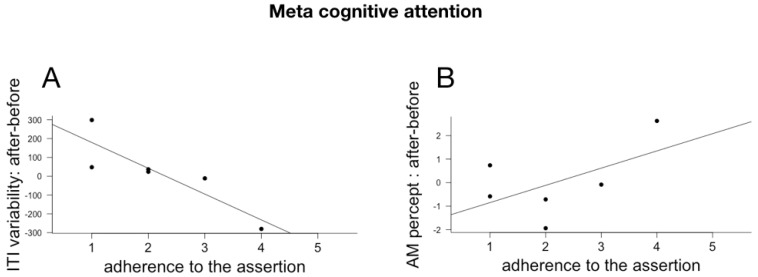
The adherence to the assertion “Sometimes the movement was so slow I did not notice changes in dancer posture” was correlated with decrease in ITI variability (−0.865, (**A**)) and increase in the report of AM percept post performance (0.6, (**B**))

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
