# Peer review of "Synchronization, Attention and Transformation: Multidimensional Exploration of the Aesthetic Experience of Contemporary Dance Spectators"

_behavsci, 2018, doi:10.3390/bs8020024_

Round 1

Reviewer 1 Report

This is a nicely written article that (1) summarizes a large background of information from dance, the psychology of time, the embodiment of mental functioning and related fields, and (2) presents a novel empirical investigation on how an audience reacts / resonates with two different dance performances. The experimental dance is based on a mindfulness body conceptualization that opens up these findings on resonance between performers and the audience to a wider context of embodiment, time, and meditation.

Here are some issues the authors can perhaps address:

-          This investigation has the character of a pilot study: (a) the number of subject per group is low which leads to difficulties in the interpretation of marginally significant findings; (b) only the Gourfink-experimental audience is equipped with psychophysiological measurements. Maybe, name this investigation pilot study.

-          The transition between the introduction and methods section is abrupt (p. 6). Just a few sentences at the end of the introduction would help to make the transition between “what is known so far” and “what we want to assess with our empirical investigation”.

-          Data processing and analysis (p. 7): “ITI longer than 3s were removed”. This applies to the internally paced tempo only? (People on average tap with 2 Hz, but at the slowest perhaps just above 1 Hz?). Make clearer that after the Procedure section you only refer to block 3 (self-paced tempo).

-          The idea of the tic-tac is nice for instructing people to tap; that may help. But I don’t believe that people would then tap either on the tic or the tic and the tac (as implied on p. 7). I have also worked with the personal tapping tempo paradigm, and this strikes me as odd. Or this then is an actually odd instruction where a mistake can happen in conveying the information.

-          Starting with last paragraph, p. 9: Some positive results are only marginally significant which is due to the low number of participants. Here the typical statistical “advise”: you could use non-parametric testing (actually indicate your tests explicitly, at some late point you once mention using Pearson’s r but not earlier); use U-Test for group differences; and Wilcoxon for post- vs. pre- within-subjects differences, and Spearman for correlations. Then one cannot do a nice 2 x 2 design; but it is more honest to the data with such a low number of people. Alternatively, you could try and transform the variables (logarithmic or square root) to make the underlying sample conform more to the assumption of normality and then do the parametric testing.

-          Last paragraph, p. 9: For ITI duration, the main effect and the interaction is mentioned but not the post-hoc effect for the Gourfink-experimental condition (which would be the target effect: after vs. before dance and vs. the control condition). Also, you can indicate the direction of difference; not just: they are different; e.g. Gourfink-experimental condition post-effect is a slowing down of the tempo.

-          The same applies to 2.1.2 indicate the direction of difference; post vs. pre performance (“stronger AM percept” and “reports of AM increased post performance” is difficult to understand).

-          “Epochè” is written like this; compare it with your writing on page 9 (epockè).

-          Fig. 3; compare A and B and adjust for aesthetic reasons …

-          The first sentence: omit the “significant change” since it is not significant.

-          Last: normally I don’t like to refer to other publications … but since it is so very much related to this research (! and inspired by the group in Paris), have a look at (just came out on Monday): Deinzer V, Clancy L, Wittmann M (2017). The sense of time while watching a dance performance. Sage Open 7, October-December 2017, 1–10.

Author Response

Dear reviewer 1,

We thank you for the careful reading of the paper and detailed and constructive feedback.

We found your comments and suggestions very useful and have done our best to integrate them in the text:

We have added (or clarified existing language) through the text notes concerning the limits imposed by the small number of subjects and the lack of physiological measures in the control group. However, we do not find it right to consider the entire study as a pilot study. For the main results concerning the changes in cognitive variables we have (what we consider) an appropriate control condition and enough subjects for a meaningful statistical comparison.

We have now reorganized the introduction (the issue was raised by all reviewers) and specifically took care to clarify the specific hypotheses and context of our study before turning to the method section

ITI: As we now clarify in the methods section, we used only the ITI from the spontaneous tempo block for analysis (in line with McCauley et al’s method)

Tic-Tac: There is no standard instruction in the literature. This specific version was proposed by a musician/engineer in our team after previous versions of the instruction (such as ‘tap according to your own tempo) were found to be opaque by participants. we added a caveat in the text. We have discussed the issue with colleagues who have extensive experience  in tapping tasks and they confirm that people sometime  perform the task on different metrical levels (in our case only on tic or on both tic and tac). In their methods, they actually half the response time of the higher metrical level but they find that eliminating these subjects,as we did, is a valid and conservative choice.

We thank you for the advice concerning the statistical tests. Indeed, in the current versions, all tests which compare individual means have been replaced with non-parametric tests. We have conserved the mixed analysis for the ITI duration and BR. We have also conserved the Pearson correlation. Given that we do not present the correlations as significative results but only as suggestive observations, we think the use of Pearson correlation is acceptable. In addition, because we correlate a 5 level questionnaire ratings, there are often ranking ambiguities that are problematic for the Spearman correlation method.

We have clarified the language concerning the directionality of the effects

We have added two references to the Deinzer et al. paper. We thank you for pointing it out to us. It is indeed very relevant.

Reviewer 2 Report

Synchronization, attention and transformation: Multidimensional exploration of the aesthetic experience of contemporary dance spectators

General remarks

The topic and the contents of the paper are quite interesting and thought-provoking. This holds true especially for the following topics: the concept of resonance, the relation between neuroaesthetics and mirroring, coupling, synchronization and coordination, the specious present, attention and vigilance, the elements of attention, such as degree of absorption, direction and object of attention. Bringing all these topics together in a coherent framework is rather difficult and this is one of the major problems of this paper. Besides there are also problems with the methodology and the global structure of the paper, which, as a whole does not meet sufficient academic standards.

Below I give first some general comments and thereafter I list up some more detailed comments.

General comments.

The global impression of this paper is that there is much interesting material. At first reading, the opening pages are very promising. The contents seem to be challenging, the list of references is impressive, and there seems to be a kind of balance between a theoretical overview and an experimental study. After careful reading, however, this positive impression evades, as there are many problems with the global structure of the paper, the style of writing and the methodology. The major problem of this manuscript is lack of coherence and not meeting academic standards. There is need of more generalizing, the reporting should be more concise and the methodology must be explained much better in order to be convincing. Here follow some general remarks.

-      The abstract is too long. There are too many details and the real take home of the paper is not stated explicitly.

-      The language use is mostly OK, but there are some minor mistakes and inconsistencies, but this is not really a problem.

-      Some concepts are introduced but are not clearly described or defined at their first appearance. Examples are the concepts of resonance, TMS, “body-mind”, ecological context, BRV, HRV, ART questionnaire

-      Some classifications are rather intuitive and are not well-motivated. They are also lacking in coherence. An example is the division of body scales (neural, physiological, sensorimotor, behavioral, experiential of affective).

-      Many parts of the main text are lacking coherence. It is difficult at times to see the wood for the trees. Many examples of individual papers are introduced, but why just these examples, and what about the others? At times, there are too many generalizations from too little material.

-      The structure of the paper is not coherent. This is most obvious by having a look at the titles and subtitles, which are lacking totally all kind of hierarchical organization. This is really very unprofessional and gives a very bad impression for the reader.

-      There are inconsistencies in the use of abbreviations. An example is the use of STP as against SMT.

-      The methodology seems to be strong but is not convincing and is not explained sufficiently clearly. There is reference to a questionnaire, but nothing is told about the questions. How were the questions collected? What was the rationale behind these questions? There is no appendix with the questions, only here and there some examples. This is really not convincing. Also, the statistics are very badly explained even with abbreviations which are very uncommon (e.g. “e.s.” What does this mean? Does this refer to “degrees of freedom”?). The number of subjects is also very low for making valuable generalizations. The figures are also very badly explained. They are not readable, the units are not explained, the figure captions are not clear and some interpretations are difficult to follow from the accompanying text.

Detailed comments

-      p.1: abstract is too long

-      p.1, last two sentences: please be consistent in the use of neuro-cognitive or neurocognitive, neuro-aesthetics or neuroaesthetics; explain the term resonance clearly at first appearance, and proceed thereafter to elaborate

-      p.2, line 3: the rationale behind the classification of body scales is not clear and not well-motivated

-      p.2, line 13: reference is missing

-      p.2: explain the abbreviation TMS; not clear for readers unfamiliar with neuroscience

-      p.2: explain the term “body-mind”; who coined this term?

-      p.2, line 8 from bottom: which kind of performance? Explain more clearly

-      p.5: references to “newness” and “alertness” are rather old. Please provide more recent updates.

-      p.6: there are inconstancies in the use of the abbreviation STP through the whole paper. SMT is used continuously.

-      p. 6: what is meant with “ecological content”? Please explain

-      p.7, 2nd alinea: three blocks; combination of “open task”: “tap as fast as you can” and the constraints of a fixed number of taps (100 taps) is very strange

-      p.8: provide at least an overview and the rationale behind the questionnaire

-      p.9: what does the abbreviation “e.s.” mean? Has this to do something with degrees of freedom? Very uncommon abbreviation.

-      p.10: figures are unreadable and are very badly explained and interpreted. Must be much clearer.

-      p.15: figure caption is not clear. Difficult to understand.

-      p. 17: use the Greek characters for epocke or italicize, is epoche not a better option?

-      p.18: motivation behind hierarchical structure/level of subtitle (ii. Direction…) is totally lacking. This holds also for the other subtitles.

-      p.19; same remark for iii. extent of …

-      p.19: meta cognitive or metacognitive? Please be consistent

-      p.20: conclusion is too short; the take home message should be more explicit here; what are the main findings and generalizations?

Given all these comments I consider this submission not to be at the needed academic standards for a journal as “Behavioral Sciences”. I suggest not to accept the paper for publication. I would suggest to the authors that they should try to make a theoretical paper and/or review paper of existing empirical studies. This could be a kind of preparation for an empirical follow-up paper. But the current paper is a bit of both, but is not really convincing.

Author Response

Dear reviewer 2,

We thank you for your careful reading of the paper and the in-depth comments. We were sorry that you found the original version of the paper not appropriate for publication and did our best to improve upon it in the current version, following your remarks.

We have shortened and streamlined the abstract

We have now explained all abbreviations and attempted to clearly introduce the different concepts such as resonance, bodymind, ecological context and resonance (though this is quite complicated in an interdisciplinary volume with potentially very diverse readership). The term body-scales has been replaced.

We have re-organized the introduction (with clearer hierarchical structure and a road map) so we hope it is easier to follow and seems more coherent (cf. also reviewr 3). We added a section to precise the role of attention in meditation practice and the effect of meditation on time perception

We have also restructured the discussion.

We have reorganized the discussion of the questionnaire. In the method section there is now a more detailed discussion of the construction of the questionnaire.  We also added the actual items in an appendix

We have made new figures with larger fonts and improved the captions

The statistical analyses have been improved and their description and discussion, as well as the discussion of the results, clarified.

We have extended the conclusion and distinguished between the significant results and the more suggestive observations.

The different typographic inconsistencies have been corrected

Reviewer 3 Report

This is a fascinating paper that will be great interest to readers of this journal. I have no major issues with the content of the paper. However, because the review and the study itself are rather complex, it is important that the discussion is presented as clearly as possible.

With this in mind, I suggest that the Authors try to clarify the headings and sections. For example, perhaps the sections should be renumbered with the introduction as 1. Then the section entitled “Slowness, continuum and duration: the choreographies of Myriam Gourfink” as section 2. The bold in text passages can then be used as subsections: 

1. Introduction

2. Slowness, continuum and duration: the choreographies of Myriam Gourfink

   2.1 Phenomenological approach to Myriam Gourfink’s work

   2.2 The making of slow movement: yoga and meditation or attention and respiration:

3. The shared present: the specious present as a window to interpersonal resonance

   3.1 Modulations of our perception in the present moment - attention and vigilance

and so on… 

I also find many small style/grammar issues: 

P.2 - From the perspective of neuroaesthetics, the phenomena of mirroring, coupling/synchronization or coordination specify the forms and levels of resonance which are at play in the reception of an art work. 

         “… that are at play...”

P.2. – Freedberg and Gallese [2] argue for a role of the mirror neuron network in art reception

          “… for the role played by the mirror…

P.2. This paragraph is now very long. Perhaps a break is needed - a new paragraph beginning with: 

        “These studies point to the crucial importance of the co-presence of bodies in the study of interpersonal resonance phenomena.”

4. P.2. “These studies point to the crucial importance of the co-presence of bodies in the study of interpersonal resonance phenomena. If the concept of the spectator as a resonator of the art work finds support in the studies relating to the phenomena of neuronal, physiological and behavioral mirroring or attunement during the reception of a work or during an interaction with others; two dimensions inherent to the notion of "resonance" seem to be marginalized in these studies: (1) The temporal extension of the resonance phenomenon and (2) The factors that condition the amplification of resonance.”  

       This passage is a little awkward: The second sentence begins with “If”, implying a “then” that never comes. 

Perhaps the new paragraph could begin with something like: 

“These studies point to the crucial importance of the co-presence of bodies in the study of interpersonal resonance phenomena - the concept of the spectator as a resonator with the  performance or art work is supported by the phenomena of neuronal, physiological, and behavioral mirroring or attunement during the reception of a work or during an interaction with others. This said, two dimensions inherent to the notion of "resonance" seem to be marginalized in these studies: (1) The temporal extension of the resonance phenomenon and (2) The factors that condition the amplification of resonance.”  

P.2.  “… have considered essentially three phenomena that could be…”

“essentially” seems awkward. Perhaps: “… have tended to focus on three phenomena that could be…”

P.2. “that can move us in our sensation, emotion, perception, taste and the way experience ourselves and the world.”

     Missing word? “… and the way we experience ourselves… ”

p.3 “…that she practices since more than twenty years.”

While this is completely understandable in English, it reads as French in style. A more ‘English’ version would be something like “… that she has been practicing for more than twenty years” 

p3. “The extreme slowness of the continuous movement induces for some spectators phenomena of inattentional blindness [36] and blindness to slow changes…"

    Missing ‘the’ – “the phenomena”

P3. “Notably the continuous observation of change as described by Gunaratana in his treatise on meditation:

“… is described by…” ? or “… observation of change is described as follows by Gunaratana in his treatise on meditation: …” ?

P.4 “For a phenomenologist, M. Gourfink description of …”

    “For a phenomenologist, M. Gourfink’s description of …”

P4. “This phenomenon of fusion of successive elements into a coherent whole was named differently "specious present", "psychological present" or "subjective present".

This is awkward. Try: “This phenomenon involving the fusion of successive elements into a coherent whole has been referred to as the "specious present", the "psychological present" or the "subjective present".

These small problems cause the reader to go back to make sure they understand the passage, interrupting the flow of the paper. There seem to be fewer of them in the discussion of the study, but see the Conclusion:

“Our results find an echo in the psychological approach to the duration of the philosopher H. Bergson [143].”

Perhaps something like:

“Our results echo the psychological approach to duration offered by the philosopher, H. Bergson [143].”

In all, I suggest that the Authors recruit a native English academic writer to carefully proof read and help tighten up the article before publication. This aside, I really enjoyed reading the paper and I am confident that it will make an excellent contribution to this journal – one that aligns closely with the theme of this special issue.

Author Response

Dear reviewer 3,

We thank you for the careful reading of our manuscript and for the encouraging and helpful feedback and suggestions.

We have now reorganized the introduction along the lines you have suggested, better reflecting its  hierarchy. We had a section to precise the role of attention in meditation practice and the effect of meditation on time perception

The discussion was also re-worked and we hope it is clearer now.

We have made all the language changes that you have suggested and had the paper proof-read by two  native speakers

Reviewer 4 Report

The article presents a study aimed at exploring the effects of a choreography involving extremely slow movements without rhythmic ruptures on the temporal perception, synchronisation, and attention of the spectators.

Strengths of the article:

I find mixed methodology adopted by the authors well-balanced, innovative yet sufficiently rigorous. Relevant theories, analysis of empirical data, and the close collaboration with a practitioner (the choreographer) all contribute to an ambitious and compelling study.

The significantly longer inter-tap intervals (ITI) reported after attending the “slow” performance compared to the “control” performance is, in my opinion, the strongest result of the study. It is supported by an adequate analysis of empirical data and constitutes a valuable finding for the research field.

Weaknesses of the article:

As clearly reported by the authors, some claims cannot be supported with significant empirical evidence due to the limited sample size.

Unfortunately, physiological data was not recorded in the control condition. This is really a shame in my opinion, as that would have allowed the authors to draw much stronger conclusions. Their claims regarding the relationship between subjective experience and physiological changes  (albeit interesting) should therefore be considered prevalently speculative.

Recommendations:

Despite its limitations, I believe the study is a valuable contribution to the interdisciplinary field of performing arts research, and may also have interesting implications for other research fields. The authors clearly reported both strengths and limitations of their study. I deeply appreciate this critical and transparent approach (which should indeed be the standard in every scientific study). However, if additional data cannot be collected at this point, particular care should be given in reporting results that have been empirically tested and assumptions that require further validation. The difference between the two should be made as clear as possible. Indeed, I can definitely see that effort has already been made in that direction, but I would recommend the authors to clearly restate in the Conclusion section which results have been attained, as well as in which direction future work should go to in order to address open questions and unverified claims. I believe this would considerably improve the already good quality of the article.

Comments on the main sections of the article:

The introduction section presents a thorough review of relevant interdisciplinary background work, including research literature on time perception, music production and perception, embodiment and interpersonal resonance, sensorimotor synchronisation, and more. The work of the choreographer that collaborated to the study is also described, giving the reader a good overview of her practice and how it relates to the themes of the study. Perhaps, the claim that properties of physical resonance apply by extension to psychological or affective resonance should be softened and/or supported by relevant references.  As it is, the claim sounds quite weak and debatable, contrary to the rest of the introduction section, which looks much more solid.

The methods section describes the procedures of the study in adequate detail, clearly reporting on design choices, tasks, analysis criteria, and also limitations (e.g. physiological data was not collected from the control group). Despite some limitations, the methodology adopted seems sound and sufficiently rigorous as well as innovative, thus it constitutes a useful contribution in itself.

The discussion section contains several references to other studies. While the works referenced are relevant, I would consider moving some of the more extensive descriptions of other studies and literature to the Introduction, at least those not immediately necessary to discuss the results. This may help maintaining the Discussion section more focused on interpreting the results of the study carried out by the authors, possibly allowing them to make their points stronger.  Also, it is usually good practice not to introduce new results in the Discussion section, so I wonder if the article would benefit from moving all the graphs and description of the data regarding the evaluation of the choreography (currently in section 3.1) to the Results section and leave the Discussion section more lightweight and to the point.

In the conclusion section, the authors could perhaps restate the main findings of the study and describe the implications such findings may have on performing arts research and practice, what new research perspectives they may lead to and how the limitations of the study may be overcome. My feeling is that the current version of the conclusion section does this only in part. Given the findings of the study, what is our new understanding of the problem investigated and outlined in the Introduction? What are the implications of the study for researchers and practitioners? And what, if any, avenues for future studies does it lead to?

Typos and other minor issues: 

P. 1: I guess received date is in 2017

TP: Is that an abbreviation for Tempo Production?

P. 2: Missing reference after “At the subjective level” (there is a [?])

P. 9: “in children []”: missing reference number

“epocke”: isn’t it spelled “epoché”?

P.14: “it will be premature to interpret…”: shouldn’t it be “it would be premature to interpret…”?

P.20 “support from the stuff”: the authors meant “staff” I suppose?

#slomo/reviews

Author Response

Dear reviewer 4,

Thank you for the careful reading of our paper and the very constructive and thoughtful comments.

We have attempted to integrate these into the new version of the article:

We have reorganized the discussion to highlight the stronger results (starting with the ITI changes) and distinguish between the statistically significant results and the more suggestive ones. We also replaced some statistical tests with non parametric versions (reviewer 1’s suggestion) which are indeed more appropriate.

We also regret not to have acquired physiological data during the control performance. As a consequence overall changes in BR after the performance cannot be interpreted as specific to the Gourfink performance.  We have clarified further this limitation in the different sections of the paper. However, it should be noted, that  we hold that the observed significant correlation between BR and AM (the degree of slowing of breath positively correlated with the degree of increase in the AM  report) is an important, novel and interpretable result which does not depend on a comparison with the control condition. Similarly, correlation between BR parameters and subjective reports are also meaningful ,albeit not statistically interpretable, without reference to a control condition.We have again re-emphasized in the new version the preliminary character of these analyses.

We have taken out the analogy with physical resonance (which was originally added for illustration only)

We have moved significant portions of the discussion into the introduction and method sections as you have suggested. We have also moved the figures pertaining to the results of the questionnaire correlations to the results section (we agree that this is the general practice. Our original choice to put them in the discussion was in order to facilitate reading).

We have reorganized and extended the conclusion section in line with your very useful suggestions

Round 2

Reviewer 2 Report

The quality of this resubmission has been improved radically compared to the first version. Most of my previous criticisms do no longer apply. The language is excellent, the style of reading is much more fluent, the overall structure of the paper is much more coherent and the technical terms are much better explained. As a whole, the paper now meets the needed academic standards.

In order to improve the paper still a little bit I list here some minor final comments and recommendations:

- page 2, line 16: “artwork” instead of “art work”?

- page 2, line 19: be consisten in using given names; either all or them or none of them (Franz Kline, Pollock, Fontana)

- page 2, line 14 from bottom: I would suggest to use the term “body-mind’ rather than “bodymind”, which seems to be a very common linguistic construction in English. This applies for all occurrences of the term in the paper.

- page 4, line 21: “non-intentional” instead of “non intentional”

- page 5, line 2 from bottom: delete capital in Succession

- page 6, line 1: “subfield” instead of “sub field”?

- page 7, line 17: delete “r” after prolongation

- page 7, subtitle 4: Use capital for first word: The shared …

- page 8, last line: “participants sat” instead of “participants set”

- page 9, line 9 and line 17 from below: be consistent in spelling in full the abbreviations ITI and ISI: “inter-tapping-intervals” as against “ Inter Stimulus Intervals”

- page 10, line 3 from below: for ease of readin it is better to place the numericals before the keywords, and not after; use lower case (i) (ii) (iii) rather than capitals (I) (II) (III)

- page 11, line 2: delete comma after “several studies”

- page 11, line 7 from below: I have some problems with the use of the term interception and exteroception here. They are used in a very loose sense. I would recommend not to use them as they refer to sensory processes rather than attentional ones; same also for later occurrences of the term

- page 12, line 12: insert blank space after [140]

- page 13, line 4: “1A”, “1B”…  instead of “1.A”?; insert blank space after “effect size”

- Figure 1: font size of upper text (Mean change in ITI…) is different

- Figure 2: It is still differet to understand these figure, as the units in the x and y axes are not explained. How are the values computed? What are the reference values. This should be explained a little more explicitly.

- Figure 3: data is a plural noun; the data “were” centered ; I do not understand “by removing the man for each group”?

- Figure 4: y axis is not readable, perhaps putting the figures below each other and enlarge their size?,

- Figure 5: enlarge size, readability of text is problematic

- Figure 6: breathing rate and rate variability seem to be at different locations in the figure and the figure caption text

- page 22, line 24: delete one of the two “not”s (not not )

- page 22, line 28: “to know whether” instead of “to know if”

- page 23, line 9: insert comma after “resonance”

- page 23, line 9 from bottom: “an online tablet task” instead of “online tablet task”

- page 26; ref. 11. “Norton series” instead of “norton series”

- take care not use again and again the sentence "to our knowledge this was the first time ..."

Author Response

Dear reviewer 2,

Thank you for the second careful reading of the manuscript. We are happy that the second version of the  manuscript seemed stronger. Your comments, as well as those of the other 3 reviewers, were extremely useful in this process. 

In the current version we have implemented all your comments and suggestions. Though bodymind was the orthography used by the referenced author that coined the term, we have replaced all instances with body-mind as it seems more standard. The terms interoception and exteroception were replaced by internally and externally oriented attention.

All plots (except of figure 3 were redrawn to improve visibility and legibility. We have also extended or improved the description of the plots in the captions (figures 2,3,5). We hope it is clearer now. Figure order of panels in figure 6 was corrected.